# Statistical learning beyond words in human neonates

**Ana Fló[1,2]\*, Lucas Benjamin[1,3,4], Marie Palu[1], Ghislaine Dehaene-Lambertz[1]**

[1]Cognitive Neuroimaging Unit, CNRS ERL 9003, INSERM U992, CEA, Université Paris Saclay, NeuroSpin center, Gif-sur-Yvette, France; [2]Department of Developmental Psychology and Socialisation and Department of Neuroscience, University of Padova, Padova, Italy; [3]Departement d'étude Cognitives, École Normale Supérieure, Paris, France; [4]Aix Marseille Univ, INSERM, INS, Inst Neurosci syst, Marseille, France

## eLife Assessment

The manuscript provides **important** new insights into the mechanisms of statistical learning in early human development, showing that statistical learning in neonates occurs robustly and is not limited to linguistic features but occurs across different domains. The evidence is **convincing** and the findings are highly relevant for researchers working in several domains, including developmental cognitive neuroscience, developmental psychology, linguistics, and speech pathology.

**\*For correspondence:**
ana.flo@unipd.it

**Competing interest:** The authors declare that no competing interests exist.

**Abstract** Interest in statistical learning in developmental studies stems from the observation that 8-month-olds were able to extract words from a monotone speech stream solely using the transition probabilities (TP) between syllables (Saffran et al., 1996). A simple mechanism was thus part of the human infant's toolbox for discovering regularities in language. Since this seminal study, observations on statistical learning capabilities have multiplied across domains and species, challenging the hypothesis of a dedicated mechanism for language acquisition. Here, we leverage the two dimensions conveyed by speech –speaker identity and phonemes– to examine (1) whether neonates can compute TPs on one dimension despite irrelevant variation on the other and (2) whether the linguistic dimension enjoys an advantage over the voice dimension. In two experiments, we exposed neonates to artificial speech streams constructed by concatenating syllables while recording EEG. The sequence had a statistical structure based either on the phonetic content, while the voices varied randomly (Experiment 1) or on voices with random phonetic content (Experiment 2). After familiarisation, neonates heard isolated duplets adhering, or not, to the structure they were familiarised with. In both experiments, we observed neural entrainment at the frequency of the regularity and distinct Event-Related Potentials (ERP) to correct and incorrect duplets, highlighting the universality of statistical learning mechanisms and suggesting it operates on virtually any dimension the input is factorised. However, only linguistic duplets elicited a specific ERP component, potentially an N400 precursor, suggesting a lexical stage triggered by phonetic regularities already at birth. These results show that, from birth, multiple input regularities can be processed in parallel and feed different higher-order networks.

## Introduction

Since speech is a continuous signal, one of the infants' first challenges during language acquisition is to break it down into smaller units, notably to be able to extract words. Parsing has been shown to rely on prosodic cues (e.g. pitch and duration changes) but also on identifying regular patterns across perceptual units. Nearly 20 years ago, *Saffran et al., 1996* demonstrated that infants are sensitive

**eLife digest** Imagine listening to a language you don't know. When does one word end, and another begin? Human infants face a similar challenge, yet remarkably, they grasp the structure of their mother tongue naturally without receiving any explicit indications. By six months, they recognize some common nouns, and by one year, they start saying their first words. This learning begins from birth, with newborns already sensitive to speech patterns.

Previous studies have shown that the likelihood of certain syllables appearing after others allows infants to detect regularity and separate speech into chunks. This is because some syllables are more predictive of what comes next than others. For example, in English, many different syllables can follow 'the'. However, it is highly likely that 'brocco' will be followed by 'li'. The ability to detect these regularities is known as statistical learning. However, whether this relies on a general mechanism or is restricted to a specific speech component, such as the sequence of syllables, remained unknown.

To investigate, Fló et al. measured brain electrical activity of newborns up to 4 days old in response to speech specifically designed to contain certain patterns of syllables or voices. In one experiment, the speech had regular patterns in the syllables, while in a second experiment, the pattern was in the voices, and each voice could utter each syllable. Unlike tracking syllable variation, which can help with learning words, voice changes within a word are unnatural and predicting them is not relevant to real-life speech processing. Therefore, if statistical learning in speech is shaped to promote language acquisition, learning should be restricted to syllable patterns. Instead, if statistical learning is a general mechanism, newborns should also detect the patterns in voice.

Analysis revealed that newborns were equally capable of discerning regular patterns in syllables despite voice changes and in voices disregarding the syllable that was pronounced. This suggests that statistical learning is a general learning mechanism that can operate across multiple features. Additionally, pseudo-words (those which resemble a real world but don't exist in the language) were presented to the newborns after they had been familiarised with speech containing either similar syllable or voice patterns. The researchers observed a specific neural response to the pseudowords only when related to syllable patterns. This neural component suggests that only syllabic structures are considered word candidates and processed by a dedicated neural network from birth.

Taken together, the findings of Fló et al. reveal insights into how humans process speech when experience with language is minimal, suggesting that statistical learning may have a broader role in early language acquisition that previously thought.

to local regularities, specifically Transitional Probabilities (TP) between syllables—that is, the probability that two particular syllables follow each other in a given sequence, $TP = P(S_{i+1}|S_i)$. In their study, 8-month-old infants were exposed to a continuous, monotonous stream of syllables composed of four randomly concatenated tri-syllabic pseudo-words. Within each word, the sequence of syllables was fixed, resulting in a TP of 1 for each syllable pair within a word. By contrast, because each word could be followed by any of the three other words, the TP for syllable pairs spanning word boundaries dropped to 1/3. The authors found that, after only 2 min of exposure to this stream, infants could distinguish between the original 'words' and 'part-words'—syllable triplets, including the between-word TP drop. Since this seminal study, statistical learning has been regarded as an essential mechanism for language acquisition because it allows for the extraction of regular patterns without prior knowledge.

During the last two decades, many studies have extended this finding by demonstrating sensitivity to statistical regularities in sequences across domains and species. For example, segmentation capacities analogous to those observed for a syllable stream are observed throughout life in the auditory modality for tones (*Kudo et al., 2011*; *Saffran et al., 1999*) and in the visual domain for shapes (*Bulf et al., 2011*; *Fiser and Aslin, 2002*; *Kirkham et al., 2002*) and actions (*Baldwin et al., 2008*; *Monroy et al., 2017*). Non-human animals, including cotton-top tamarins (*Hauser et al., 2001*), rats (*Toro and Trobalón, 2005*), dogs (*Boros et al., 2021*), and chicks (*Santolin et al., 2016*), have also been shown to be sensitive to TPs between successive events. While the level of complexity that each species can track might differ, statistical learning appears as a general learning mechanism for auditory and visual sequence processing (for a review of statistical learning capacities across species, see *Santolin and Saffran, 2018*).

Using near-infra-red spectroscopy (NIRS) and electroencephalography (EEG), we have shown that statistical learning is observed in sleeping neonates (*Fló et al., 2022a*; *Fló et al., 2019*), highlighting the automaticity of this mechanism. We also discovered that tracking statistical probabilities might not lead to stream segmentation in the case of quadrisyllabic words in both neonates and adults, revealing an unsuspected limitation of this mechanism (*Benjamin et al., 2023*). Here, we aimed to further characterise this mechanism to shed light on its role in the early stages of language acquisition. Specifically, we addressed two questions: First, we investigated whether statistical learning in human neonates is a general learning mechanism applicable to any speech feature or whether there is a bias in favour of computations on linguistic content to extract words. Second, we explored the level at which newborns compute transitions between syllables, at a low auditory level (i.e. between the presented events) or at a later phonetic level, after normalisation through irrelevant dimensions such as voices.

To test this, we have taken advantage of the fact that syllables convey two important pieces of information for humans: what is being said and who is speaking, that is linguistic content and speaker's identity. While statistical learning can be helpful to word extraction, a statistical relationship between successive voices is not of obvious use and could even hinder word extraction if instances of a word uttered by a different speaker are considered independently. However, as auditory processing is organised along several hierarchical and parallel pathways integrating different spectro-temporal dimensions (*Belin et al., 2000*; *DeWitt and Rauschecker, 2012*; *Norman-Haignere et al., 2015*; *Zatorre and Belin, 2001*), statistical learning might be computed on one dimension independently of the variation of the other along the linguistic and the voice pathways in parallel. Numerous behavioural and brain imaging studies have revealed phonetic normalisation across speakers in infants (*Dehaene-Lambertz and Pena, 2001*; *Gennari et al., 2021*; *Kuhl and Miller, 1982*). By the second half of the first year, statistical learning has also been shown to occur even when different voices are used (*Estes and Lew-Williams, 2015*) or when natural speech is presented, where syllable production can vary from instance to instance (*Hay et al., 2011*; *Pelucchi et al., 2009*). Therefore, we hypothesised that neonates would compute TPs between syllables even when each syllable is produced by a different speaker, relying on a normalisation process at the syllable or phonetic level. However, our predictions regarding TPs learning across different voices were more open-ended. Either statistical learning is universal and can be similarly computed over any feature comprising voices, or listening to a speech stream favours the processing of phonetic regularities over other non-linguistic dimensions of speech and thus hinders the possibility of computing regularities over voices.

To study these possibilities, we constructed artificial streams using six consonant-vowel (CV) syllables produced by six voices (*Appendix 1—table 1*, *Appendix 1—table 2*), resulting in 36 possible tokens (6 syllables ×6 voices). To form the streams, tokens were combined either by imposing structure to their phonetic content (Experiment 1: Structure over Phonemes) or their voice content (Experiment 2: Structure over Voices), while the second dimension varied randomly (*Figure 1*). The structure consisted of the random concatenation of three duplets (i.e. two-syllable units) defined only by one of the two dimensions. For example, in Experiment 1, one duplet could be *petu* with each syllable uttered by a random voice each time they appear in the stream (e.g. *pe* is produced by voice[1] and *tu* by voice[6] in one instance and in another instance *pe* is produced by voice[3] and *tu* by voice[2]). In contrast, in Experiment 2, one duplet could be the combination [voice[1]- voice[6]], each uttering randomly any of the syllables.

If infants at birth compute regularities based on a neural representation of the syllable as a whole, that is comprising both phonetic and voice content, this would require computing a 36×36 TPs matrix relating each token. Under this computation, TPs in the structured stream would alternate between 1/6 within words and 1/12 between words. We predicted infants would fail the task in both experiments as previous studies showing successful segmentation in infants commonly use higher within-word TPs (usually 1) and far fewer tokens (typically 4–12) (*Saffran and Kirkham, 2018*). By contrast, if speech input is processed along the two studied dimensions in distinct pathways, it enables the calculation of two independent TP matrices of 6×6 between the six voices in a voice pathway and between the six syllables in a phonetic pathway. These computations would result in TPs alternating between 1 and 1/2 for the informative feature while remining uniform at 1/5 for the uninformative feature, leading to stream segmentation based on the informative dimension.

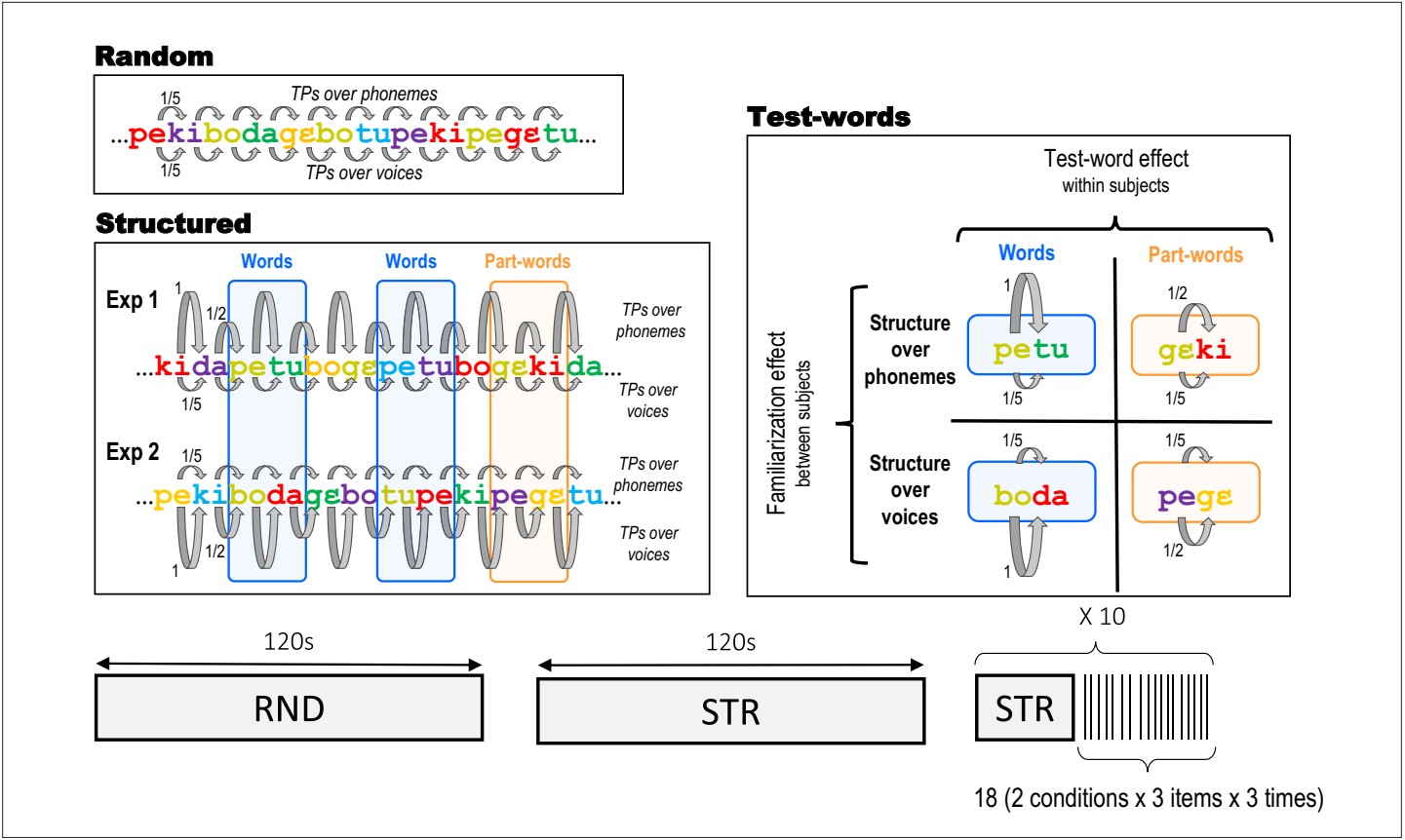

**Figure 1.** Experimental protocol. The experiments started with a Random stream (120 s) in which both syllables and voices changed randomly, followed by a long-Structured stream (120 s). Then, 10 short familiarisation streams (30 s), each followed by test blocks comprising 18 isolated duplets (SOA 2–2.3 s) were presented. Example streams are presented to illustrate the construction of the streams, with different colours representing different voices. In Experiment 1, the Structured stream had a statistical structure based on phonemes (TPs alternated between 1 and 0.5), while the voices were randomly changing (uniform TPs of 0.2). For example, the two syllables of the word 'petu' were produced by different voices, which randomly changed at each presentation of the word (e.g. 'yellow' voice and 'green' voice for the first instance, 'blue' and 'purple' voice for the second instance, etc..). In Experiment 2, the statistical structure was based on voices (TPs alternated between 1 and 0.5), while the syllables changed randomly (uniform TPs of 0.2). For example, the 'green' voice was always followed by the 'red' voice, but they were randomly saying different syllables 'boda' in the first instance, 'tupe' in the second instance, etc... The test duplets in the recognition test phase were either Words (TP = 1) or Partwords (TP = 0.5). Words and Partwords were defined in terms of phonetic content for Experiment 1 and voice content for Experiment 2.

As in our previous experiments (*Benjamin et al., 2023*; *Fló et al., 2022b*), we used high-density EEG (128 electrodes) to study speech segmentation abilities. Using artificial language with syllables with a fixed duration elicits Steady State Evoked Potentials (SSEP) at the syllable rate. Crucially, if the artificial language presents a regular structure (i.e. regular drops in TPs marking word boundaries) and if the structure is perceived, then the neural responses reflect the slower frequency of the word as well (*Buiatti et al., 2009*). In other words, the brain activity becomes phase-locked to the regular input, increasing the Inter Trial Coherence (ITC) and power at the input regularity frequencies. Under these circumstances, the analysis in the frequency domain is advantageous since fast and periodic responses can be easily investigated by looking at the target frequencies without considering their specific timing (*Kabdebon et al., 2022*). The phenomenon is also named frequency tagging or neural entrainment in the literature. Here, we will refer to it indistinctively as SSEP or neural entrainment since we do not aim to make any hypothesis on the origin of the response (i.e. pure evoked response or phase reset of endogenous oscillations, *Giraud and Poeppel, 2012*).

Our study used an orthogonal design across two groups of 1- to 4-day-old neonates. In Experiment 1 (34 infants), the regularities in the speech stream were based on the phonetic content, while the voices varied randomly (Phoneme group). Conversely, in Experiment 2 (33 infants), regularities were based on voices, while the phonemes changed randomly (Voice group). Both experiments started with

a control stream in which both features varied randomly (i.e. Random stream, 120 s). Next, neonates were exposed to the Structured stream (120 s) with statistical structure over one or the other feature. The experiments ended with ten sets of 18 test duplets presented in isolation, preceded by short Structured streams (30 s) to maintain learning (*Figure 1*). Half of the test duplets corresponded to familiar regularities (Words, TP = 1), and the other half were duplets present in the stream but which straddled a drop in TPs (Partwords, TP = 0.5).

To investigate online learning, we quantified the ITC as a measure of neural entrainment at the syllable (4 Hz) and word rate (2 Hz) during the presentation of the continuous streams. For the recognition process, we compared ERPs to Word and Part-Word duplets. We also tested 57 adult participants in a comparable behavioural experiment to investigate adults' segmentation capacities under the same conditions.

## Results
### Neural entrainment during the familiarisation phase

To measure neural entrainment, we quantified the ITC in non-overlapping epochs of 7.5 s. We compared the studied frequency (syllabic rate 4 Hz or duplet rate 2 Hz) with the 12 adjacent frequency bins following the same methodology as in our previous studies.

For the Random streams, we observed significant entrainment at syllable rate (4 Hz) over a broad set of electrodes in both experiments ($p<0.05$, FDR corrected) and no enhanced activity at the duplet rate for any electrode ($p>0.05$, FDR corrected). Concerning the Structured streams, ITC increased at both the syllable and duplet rate ($p<0.05$, FDR corrected) in both experiments (*Figure 2A and B*). The duplet effect was localised over occipital and central-left electrodes in the Phoneme group and over occipital and temporal-right electrodes in the Voice group.

We also directly compared the ITC at both frequencies of interest between the Random and Structured conditions (*Figure 2C*). We found electrodes with significantly higher ITC at the duplet rate during Structured streams than Random streams in both experiments ($p<0.05$, FDR corrected). We also found electrodes with higher entrainment at syllable rate during the Structured than Random streams in both experiments ($p<0.05$, FDR corrected). This effect might result from stronger or more phase-locked responses to syllables when the input is structured. Since the first harmonic of the duplet rate (2x2 Hz) coincides with the syllable rate (4 Hz), word entrainment during the structured streams could also contribute to this effect. However, this contribution is unlikely since the electrodes showing higher 4 Hz entrainment during Structured than Random streams differ from those showing duplet-rate activity at 2 Hz.

Finally, we looked for an interaction effect between groups and conditions (Structured vs. Random streams; *Figure 2C*). A few electrodes show differential responses between groups, reflecting the topographical differences observed in the previous analysis, notably the trend for stronger ITC at 2 Hz over the left central electrodes for the Phoneme group compared to the Voice group, but none survive multiple comparison corrections.

### Learning time-course

To investigate the time course of the learning, we computed neural entrainment at the duplet rate in sliding time windows of 2 min with a 1 s step across both random and structured streams (*Figure 2D*). Notice that because the integration window was 2 min long, the entrainment during the first minute of the Structured stream included data from the random stream. To test whether ITC at 2 Hz increased during long Structured familiarisation (120 s), we fitted a Linear Mixed Model (LMM) with a fixed effect of time and random slopes and interceptions for individual subjects: $ITC \sim -1 + time + (1 + time|subject)$. In the Phoneme group, we found a significant time effect ($\beta=4.16 \times 10^{-3}$, 95% CI=[$2.06\times10^{-3}$, $6.29\times10^{-3}$], SE = $1.05 \times 10^{-3}$, $p=4 \times 10^{-4}$), as well as in the Voice group ($\beta=2.46 \times 10^{-3}$, 95% CI=[$2.6\times10^{-4}$, $4.66\times10^{-3}$], SE = $1.09 \times 10^{-3}$, $p=0.03$). To test for differences in the time effect between groups, we included all data in a single LMM: $ITC \sim -1 + time * group + (1 + time|subject)$. The model showed a significant fixed effect of time for the Phoneme group consistent with the previous results ($\beta=4.22 \times 10^{-3}$, 95% CI=[$1.07\times10^{-3}$, $7.37\times10^{-3}$], SE = $1.58 \times 10^{-3}$, $p=0.0096$), while the fixed effect estimating the difference between the Phoneme and Voice groups was not significant ($\beta=-1.84 \times 10^{-3}$, 95% CI=[$-6.29\times10^{-3}$, $2.61\times10^{-3}$], SE = $2.24 \times 10^{-3}$, $p=0.4$).

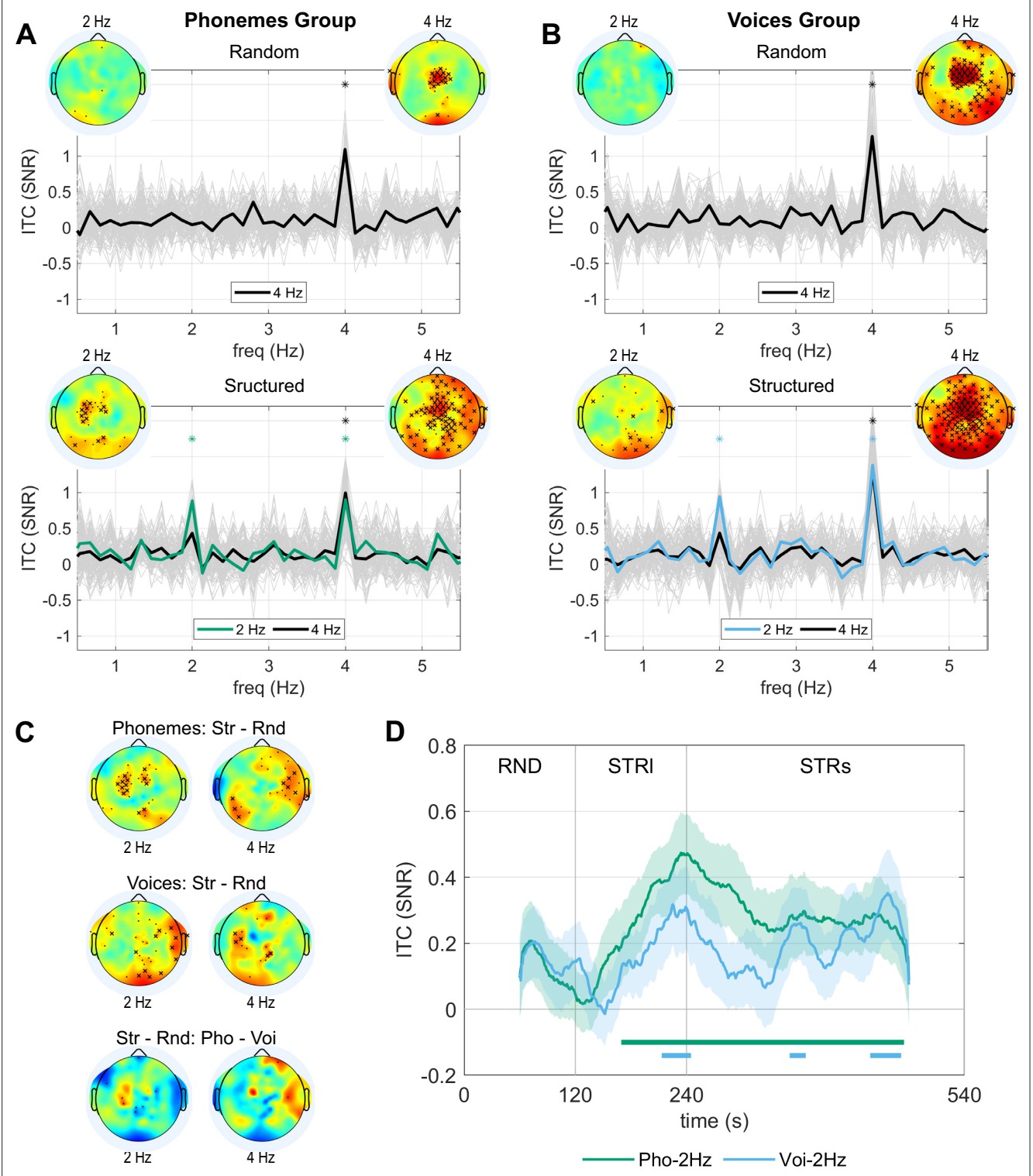

**Figure 2.** Neural entrainment during the random and structured streams. (**A**) SNR for the ITC during the Random and Structured streams of Experiment 1 (structure on phonetic content) (n=32). The topographies represent the entrainment in the electrode space at the syllabic (4 Hz) and duplet rates (2 Hz). Crosses indicate the electrodes showing enhanced neural entrainment (cross: p<0.05, one-sided paired permutation test, FDR corrected by the number of electrodes; dot: p<0.05, without FDR correction). Colour scale limits [–1.8, 1.8]. The entrainment for each electrode is shown in light grey. The thick black line shows the mean over the electrodes with significant entrainment relative to the adjacent frequency bins at the syllabic rate (4 Hz) (p<0.05 FDR corrected). The thick green line shows the mean over the electrodes showing significant entrainment relative to the adjacent frequency bins at the duplet rate (2 Hz) (p<0.05 FDR corrected). The asterisks indicate frequency bins with entrainment significantly higher than on adjacent frequency

*Figure 2 continued on next page*

*Figure 2 continued*

bins for the average across electrodes (p<0.05, one-sided permutation test, FDR corrected for the number of frequency bins). (**B**) Analog to A for Experiment 2 (structure on voice content) (n=32). (**C**) The first two rows show the topographies for the difference in entrainment during the Structured and Random streams at 4 Hz and 2 Hz for both experiments. Crosses indicate the electrodes showing stronger entrainment during the Structured stream (cross: p<0.05, one-sided paired permutation test, FDR corrected by the number of electrodes; dot: p<0.05, without FDR correction). The bottom row shows the interaction effect by comparing the difference in entrainment during the Structured and Random streams between Experiments 1 and 2. Crosses indicate significant differences (cross: p<0.05, two-sided unpaired permutation test, FDR corrected by the number of electrodes; dot: p<0.05, without FDR correction). (**D**) Time course of the neural entrainment at 4 Hz for the average over electrodes showing significant entrainment during the Random stream and at 2 Hz for the average over electrodes showing significant entrainment during the Structured stream (Phoneme: green line, Voice blue line). The shaded area represents standard errors. The horizontal lines on the bottom indicate when the entrainment was larger than 0 (p<0.05, one-sided t-test, corrected by FDR by the number of time points).

## ERPs during the test phase

To test the recognition process, we also measured ERP to isolated duplets afterwards. The average ERP to all conditions merged is shown in *Figure 3—figure supplement 1*. We investigated (1) the main effect of test duplets (Word vs. Part-word) across both experiments, (2) the main effect of familiarisation structure (Phoneme group vs. Voice group), and finally (3) the interaction between these two factors. We used non-parametric cluster-based permutation analyses (i.e. without a priori ROIs; *Oostenveld et al., 2011*).

The difference between Word and Part-word consisted of a dipole with a median positivity and a left temporal negativity ranging from 400 to 1500ms, with a maximum around 800–900ms (*Figure 3*). Cluster-based permutations recovered two significant clusters around 500–1500ms: a frontal-right positive cluster (p=0.019) and a left temporal negative cluster (p=0.0056). A difference between groups was observed consisting of a dipole that started with a right temporal positivity left temporo-occipital negativity around 300ms and rotated anti-clockwise to bring the positivity over the frontal electrodes and the negativity at the back of the head (500–800ms; *Figure 3—figure supplement 2*). Cluster-based permutations on the Phoneme group vs. Voice group recovered a posterior cluster (p=0.018) around 500ms; with no positive cluster reaching significance (p>0.10). A cluster-based permutation analysis on the interaction effect, that is comparing Words - Part-Words between both experiments, showed no significant clusters (p>0.1).

As cluster-based statistics are not very sensitive, we also analysed the ERPs over seven ROIS defined on the grand average ERP of all merged conditions (see Methods). Results replicated what we observed with the cluster-based permutation analysis with similar differences between Words and Part-words for the effect of familiarisation and no significant interactions. Results are presented in SI. The temporal progression of voltage topographies for all ERPs is presented in *Figure 3—figure supplement 2*. To verify that the effects were not driven by one group per duplet type condition, we ran a mixed two-way ANOVA for the average activity in each ROI and significant time window, with duplet type (Word/Part-word) as within-subjects factor and familiarisation as between-subjects factor. We did not observe significant interactions in any case. Future studies should consider a within-subject design to gain sensitivity to possible interaction effects.

## Adult's behavioural performance in the same task

Adult participants heard a Structure Learning stream lasting 120 s and then ten sets of 18 test duplets preceded by Short Structure streams (30 s). For each test duplet, they had to rate its familiarity on a scale from 1 to 6. For the group familiarised with the Phoneme structure, there was a significant difference between the scores attributed to Words and Part-words ($t(26)=2.92$, $p=0.007$, Cohen's $d=0.562$). The difference was marginally significant for the group familiarised with the Voice structure ($t(29)=2.0443$, $p=0.050$, Cohen's $d=0.373$; *Figure 4*). A 2-way ANOVA with test-duplets and familiarisation as factors revealed a main effect of Word ($F(1,55)=12.52$, $p=0.0008$, $\eta_g^2=0.039$), no effect of familiarisation ($F(1,55) <1$), and a significant interaction Word ×Familiarisation ($F(1,55) = 5.28$, $p=0.025$, $\eta_g^2=0.017$).

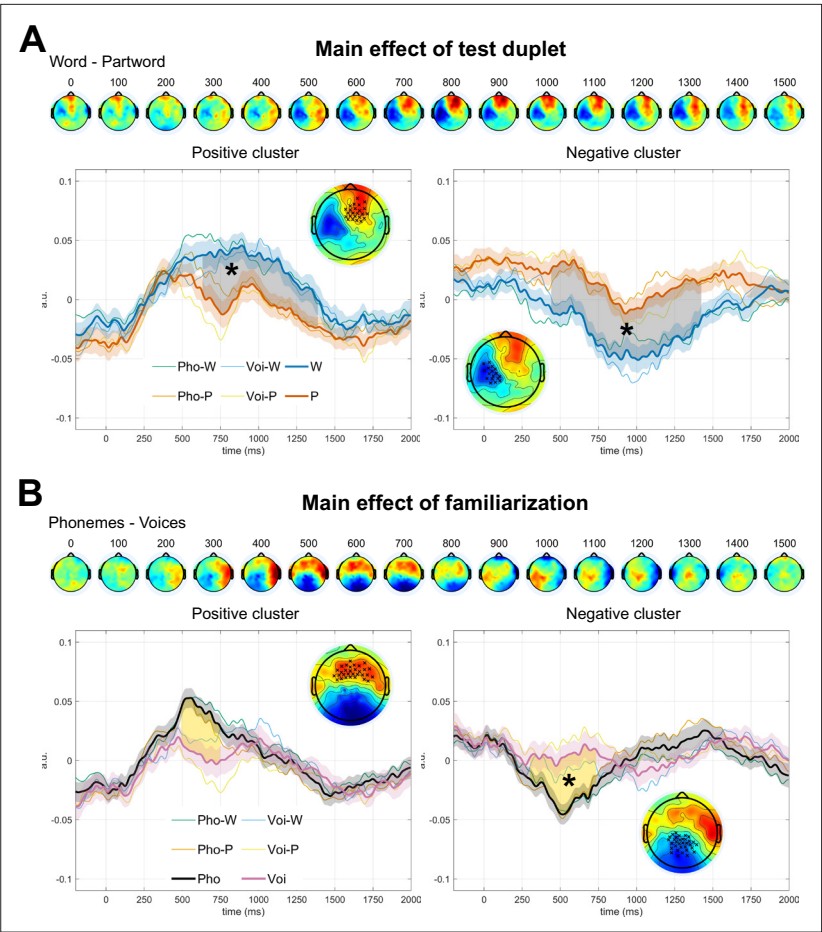

**Figure 3.** Cluster-based permutation analysis of ERPs to isolated duplets during recognition The topographies show the difference between the two conditions corresponding to each main effect. Results obtained from the cluster-based permutation analyses are shown at the bottom of each panel. Thick lines correspond to the grand averages for the two main tested conditions. Shaded areas correspond to the standard error across participants. Thin lines show the ERPs separated by duplet type and familiarisation type. The shaded areas between the thick lines show the time extension of the cluster. The topographies correspond to the difference between conditions during the time extension of the cluster. The electrodes belonging to the cluster are marked with a cross. Significant clusters are indicated with an asterisk. Color scale limits [–0.07, 0.07] a.u. (A) Main effect of Test-duplets (Words - Part-words) over a frontal-right positive cluster (p=0.019) and a left temporal negative cluster (p=0.0056) (n = 67 Words, n = 67 Part-words). (B) Main effect of familiarisation (Phonemes - Voices) over a posterior negative cluster (p=0.018) (n = 68 Phonemes, n = 66 Voices). The frontal positive cluster did not reach significance (p=0.12). Results are highly comparable to the ROIs-based analysis (*Figure 3—figure supplement 1* and *Figure 3—figure supplement 3*).

The online version of this article includes the following figure supplement(s) for figure 3:

**Figure supplement 1.** Topographies for the grand average ERP ERP across all participants for both Experiments.

**Figure supplement 2.** Topographies for the ERPs to test words during recognition Color scale limits [–0.07, 0.07] a.u.

**Figure supplement 3.** Result for the ROI analysis of the ERPs to test words during recognition.

# Discussion
## Statistical learning is a general learning mechanism

In two experiments, we compared auditory statistical learning over a linguistic and a non-linguistic dimension in sleeping neonates. We took advantage of the possibility of constructing streams based on the same 36 tokens (6 syllables ×6 voices), the only difference between the experiments being the arrangement of the tokens in the streams. We showed that neonates were sensitive to regularities

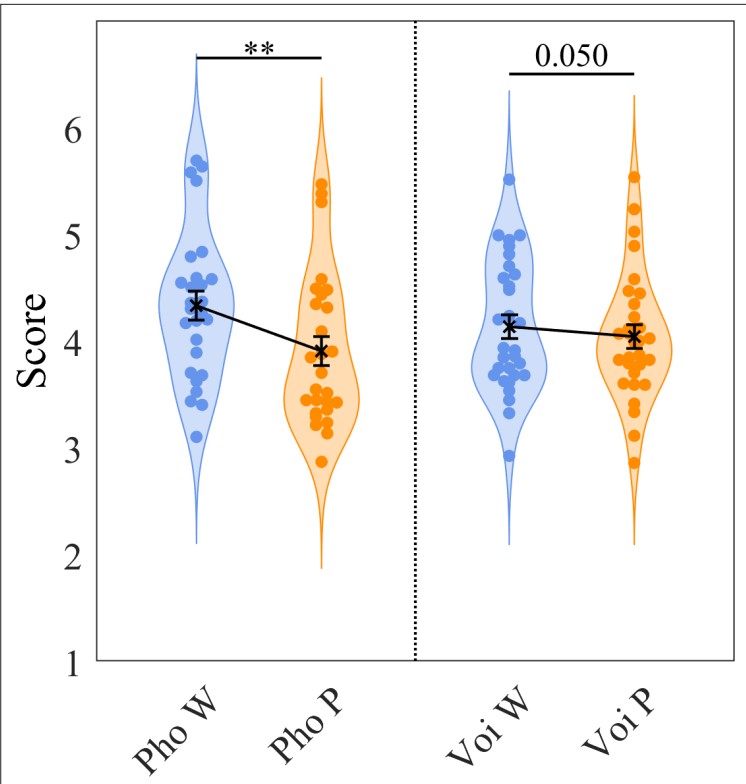

**Figure 4.** Adults' behavioural experiment. Each subject's average score attributed to the Words (blue) and Partwords (orange) is represented. On the right, for the group familiarised with the Phoneme structure (n=27) and on the left, for the group familiarised with the Voice structure (n=30). The difference between test duplets was significant for the Phoneme group (p=*0.007*) and only marginally significant for the Voice group (p=*0.050*). There was also a significant interaction group ×duplet type (p=0.025).

based either on the phonetic or the voice dimensions of speech, even in the presence of a non-informative feature that must be disregarded.

Parsing based on statistical information was revealed by steady-state evoked potentials at the duplet rate observed around 2 min after the onset of the familiarisation stream and by different ERPs to Words and Part-words presented during a recognition phase in both experiments. Despite variations in the other dimension, statistical learning was possible, showing that this mechanism operates at a stage when these dimensions have already been separated along different processing pathways. Our results, thus, revealed that linguistic content and voice identity are calculated independently and in parallel. This result confirms that, even in newborns, the syllable is not a holistic unit (*Gennari et al., 2021*) but that the rich temporo-frequential spectrum of speech is processed in parallel along different networks, probably using different integration factors (*Boemio et al., 2005*; *Moerel et al., 2012*; *Zatorre and Belin, 2001*). While statistical learning has already been described in many domains in neonates and young infants (*Fló et al., 2019*; *Fló et al., 2022b*; *Kirkham et al., 2002*; *Kudo et al., 2011*), we add here that even sleeping neonates distinctly applied it to potentially all dimensions in which speech is factorised in the auditory cortex (*Gennari et al., 2021*; *Gwilliams et al., 2022*). Our result contribute to a growing body of evidence supporting the universality of statistical learning —albeit with potential quantitative differences (*Frost et al., 2015*; *Ren and Wang, 2023*; *Santolin and Saffran, 2018*). This mechanism might be rooted in associative learning processes relying on the co-existence of event representations driven by slow activation decays (*Benjamin et al., 2024*).

We observed no clear processing advantage for the linguistic dimension over the voice dimension in neonates. The ability to track regularities in parallel for different speech features provides newborns with a powerful tool to create associations between recurring events and uncover structure. Future work could explore whether they can simultaneously track multiple regularities.

## Differences between statistical learning over voices and over phonemes

While the main pattern of results between experiments was comparable, we did observe some differences. The word-rate steady-state response (2 Hz) for the group of infants exposed to structure over phonemes was over posterior electrodes and left lateralised over central electrodes, while the group of infants hearing structure over voices showed mostly entrainment over posterior and right temporal electrodes. Auditory ERPs, after reference-averaged, typically consist of a central positivity and posterior negativity. These results are consistent with statistical learning in distinct lateralised neural networks to process speech's phonetic and voice content. Recent brain imaging studies in infants do indeed show hemispheric biases in precursors of later networks (*Blasi et al., 2011*; *Dehaene-Lambertz et al., 2010*; *Mahmoudzadeh et al., 2013*), even if specialisation increases during development (*Shultz et al., 2014*; *Sylvester et al., 2023*). However, the hemispheric differences reported here should be considered cautiously since the group comparison did not survive multiple comparison corrections. Future work investigating the neural networks involved should implement a within-subject design to gain statistical power.

The time course of entrainment at the duplet rate revealed that it emerged at a similar time for both statistical structures. While the duplet rate response seemed more stable in the Phoneme group (evidenced by a sustained ITC at the word rate above zero and a steeper slope of increase), no significant difference was observed between the groups. Furthermore, since both groups exhibited differences in the Words vs Part-words comparison during the recognition phase, it is unlikely that the differences observed during exposure to the stream were due to poorer computation of statistical transitions for voice regularities compared to phoneme regularities. Another explanation may lie not in the statistical calculation but in the stability of the voice representation itself, as voice processing challenges have been reported in both children and adults (*Johnson et al., 2011*; *Mahmoudzadeh et al., 2016*). This could disrupt stable chunking during the stream even while preserving word recognition during the test. In a previous study, we observed that entrainment failed to emerge under certain conditions despite neonates successfully computing TPs—likely due to the absence of chunking (*Benjamin et al., 2023*).

## Phoneme regularities might trigger a lexical search

In the test phase using isolated duplets, we observed a significant difference between groups: A dipole, consisting of a posterior negative pole and a frontal positivity, was recorded around 500ms following linguistic duplets but not voice duplets. Since the acoustic properties of the duplets were identical in both experiments, the difference can only be attributed to an endogenous process modulating duplet processing driven by the type of regularity to which neonates were exposed during the familiarisation phase.

Given its topography, latency and the phonetic context in which it appears, this component might be a precursor of the N400, a negative deflection at 200–600ms over central-parietal electrodes elicited by lexico-semantic manipulations in adults (*Kutas and Federmeier, 2011*). Previous studies have already postulated components analogous to the N400 in infants. For example, a posterior negativity has been reported in infants as young as five months when hearing their own name compared to a stranger's name (*Parise et al., 2010*), when a pseudo-word was consistently vs inconsistently associated with an object (*Friedrich and Friederici, 2011*), and when they saw unexpected vs expected actions (*Reid et al., 2009*). These results suggest that such negativity might be related to semantic processing. As is often the case in infants, the latency of the component was delayed, and its topography was more posterior compared to older developmental stages (*Friedrich and Friederici, 2005*; *Junge et al., 2021*).

Furthermore, even in the absence of clear semantic content, an N400 has been reported in adults listening to artificial languages. For instance, *Sanders et al., 2002* observed an N400 in adults listening to an artificial language after prior exposure to isolated pseudo-words. Other studies have demonstrated larger N400 amplitudes in adults when listening to structured streams compared to random sequences, whether these sequences consisted of syllables (*Cunillera et al., 2006*; *Cunillera et al., 2009*), tones (*Abla et al., 2008*), or shapes (*Abla and Okanoya, 2009*). Comparing ERPs from the recognition phase with those elicited by duplets during the familiarisation stream was not feasible due to the weaker signal-to-noise ratio in continuous stream recordings versus isolated

stimuli and baseline challenges specific to infant background EEG (*Eisermann et al., 2013*). However, some inferences can still be drawn regarding the differences between the phoneme and voice experiments. Neural responses in both the learning and recognition phases could reflect a top-down effect: Phonetic regularities, but not voice regularities, would induce a lexical search during the presentation of the isolated duplets or at least activate a proto-lexical network.

Previous evidence suggests that infants extract and store possible word forms even before associating them with a clear meaning (*Jusczyk and Hohne, 1997*). For example, stronger fMRI activation for forward speech than backward speech in the left angular gyrus in 3-month-old infants has been linked to the activations of possible word forms in a proto-lexicon for native language sentences (*Dehaene-Lambertz et al., 2002*). Similarly, *Shukla et al., 2011* demonstrated that chunks extracted from the speech stream serve as candidate words to which meanings can be attached. In their study, 6-month-olds spontaneously associated a pseudo-word extracted from natural sentences with a visual object. Although their experiment relied on TP and prosodic cues to extract the word, while our study only used statistical cues, this spontaneous bias to treat possible word forms as referring to a meaning (see also *Bergelson and Aslin, 2017*) might trigger activation along a lexicon pathway, explaining the difference between the two experiments: Only speech chunks based on phonetic regularities, and not voice regularities, are viable word candidates.

A lexical entry might also explain the more sustained activity during the familiarisation stream in the phonemes group, as the chunk might be encoded as a putative word in this admittedly rudimentary but present lexical store. In this hypothesis, the neural entrainment may reflect not only TPs but also the recovery of the 'lexical' item.

## Adults also learn voice regularities

Finally, we would like to emphasise that it is highly unnatural for a word not to be produced by the same speaker, nor for speakers to exhibit the kind of statistical relationships used here. Despite several years of exposure to speech, adults still demonstrated some learning of voice duplets, supporting the hypothesis of a general and automatic statistical learning ability. However, they also clearly displayed an advantage for phonetic regularities over voice regularities, revealed by the significant interaction Familiarisation ×Word, not observed in neonates.

This difference might have several not mutually exclusive explanations. First, it may stem from the behavioural test being a more explicit measure of word recognition than the implicit task allowed by EEG recordings. Second, adults may perform better with phoneme structure due to a more effective auditory normalisation process or the additional use of a writing code for phonemes, which does not exist for voices. Finally, neonates, having little experience and therefore fewer expectations or constraints, might serve as better revealers of the possibilities afforded by statistical learning compared to older participants.

## Conclusion

Altogether, our results show that statistical learning works similarly on different speech features in human neonates with no clear advantage for computing linguistically relevant regularities in speech. This supports the idea that statistical learning is a general learning mechanism, probably operating on common computational principles across neural networks (*Benjamin et al., 2024*) but within different neural networks with a different chain of operations: phonetic regularities induce a supplementary component –not seen in the case of voice regularities—that we related to a lexical N400. Understanding how statistical learning computations over linguistically relevant dimensions, such as the phonetic content of speech, are extracted and passed on to subsequent processing stages might be fundamental to uncovering how the infant brain acquires language. Further research is needed to understand how extracting regularities over different features articulates the language network.

## Methods
### Participants

Participants were healthy-full-term neonates with normal pregnancy and birth (GA >38 weeks, Apgar scores ≥7/8 at 1/5 min, birthweight >2.5 Kg, cranial perimeter ≥33.0 cm), tested at the Port Royal Maternity (AP-HP), in Paris, France. Parents provided informed consent. The regional ethical

committee for biomedical research (Comité de Protection des Personnes Region Centre Ouest 1, EudraCT/ID RCB: 2017-A00513-50) approved the protocol, and the study was carried out according to relevant guidelines and regulations. 67 participants (34 in Experiment 1 and 33 in Experiment 2) who provided enough data without motion artefacts were included (Experiment 1: 19 females; 1–4 days old; mean GA: 39.3 weeks; mean weight: 3387 g; Experiment 2: 15 females; 1–4 days old; mean GA: 39.0 weeks; mean weight: 3363 g). 12 other infants were excluded from the analyses (11 due to fussiness; 1 due to bad data quality).

## Stimuli

The stimuli were synthesised using the MBROLA diphone database (*Dutoit et al., 1996*). Syllables had a consonant-vowel structure and lasted 250ms (consonants 90ms, vowels 160ms). Six different syllables (*ki, da, pe, tu, bo, gɛ*) and six different voices were used (*fr3, fr1, fr7, fr2, it4, fr4*), resulting in a total of 36 syllable-voice combinations, from now on, tokens. The voices could be female or male and have three different pitch levels (low, middle, and high) (*Appendix 1—table 1*). We performed post-hoc tests to ensure that the results were not driven by a perception of two voices: female and male (see Appendix). The 36 tokens were synthesised independently in MBROLA, their intensity was normalised, and the first and last 5ms were ramped to zero to avoid 'clicks'. The streams were synthesised by concatenating the tokens' audio files, and they were ramped up and down during the first and last 5 s to avoid the start and end of the stream serving as perceptual anchors.

The Structured streams were created by concatenating the tokens in such a way that they resulted in a semi-random concatenation of the duplets (i.e. pseudo-words) formed by one of the features (syllable/voice) while the other feature (voice/syllable) vary semi-randomly. In other words, in Experiment 1, the order of the tokens was such that Transitional Probabilities (TPs) between syllables alternated between 1 (within duplets) and 0.5 (between duplets), while between voices, TPs were uniformly 0.2. The design was orthogonal for the Structured streams of Experiment 2 (i.e. TPs between voices alternated between 1 and 0.5, while between syllables were evenly 0.2). The random streams were created by semi-randomly concatenating the 36 tokens to achieve uniform TPs equal to 0.2 over both features. The semi-random concatenation implied that the same element could not appear twice in a row, and the same two elements could not repeatedly alternate more than two times (i.e. the sequence $X_k X_j X_k X_j$, where $X_k$ and $X_j$ are two elements, was forbidden). Notice that with an element, we refer to a duplet when it concerns the choice of the structured feature and to the identity of the second feature when it involves the other feature. The same statistical structures were used for both Experiments, only changing over which dimension the structure was applied. The learning stream lasted 120 s, with each duplet appearing 80 times. The 10 short structured streams lasted 30 s each, each duplet appearing a total of 200 times (10×20). The same random stream was used for both Experiments, and it lasted 120 s.

In Experiment 1, the duplets were created to prevent specific phonetic features from facilitating stream segmentation. In each experiment, two different structured streams (lists A and B) were used by modifying how the syllables/voices were combined to form the duplets (*Appendix 1—table 2*). Crucially, the Words/duplets of list A are the Part-words of list B and vice versa any difference between those two conditions can thus not be caused by acoustical differences. Participants were randomly assigned and balanced between lists and Experiments.

The test words were duplets formed by the concatenation of two tokens, such that they formed a Word or a Part-word according to the structured feature.

## Procedure and data acquisition

Scalp electrophysiological activity was recorded using a 128-electrode net (Electrical Geodesics, Inc) referred to the vertex with a sampling frequency of 250 Hz. Neonates were tested in a soundproof booth while sleeping or during quiet rest. The study involved: (1) 120 s of a random stream, (2) 120 s of a structured stream, (3) 10 series of 30 s of structured streams followed by 18 test sequences (SOA 2–2.3 s).

## Data pre-processing

Data were band-pass filtered 0.1–40 Hz and pre-processed using custom MATLAB scripts based on the EEGLAB toolbox 2021.0 (*Delorme and Makeig, 2004*) according to the APICE pre-processing pipeline to recover as much free-artifacts data as possible (*Fló et al., 2022b*).

## Neural entrainment

The pre-processed data were further high-pass filtered at 0.2 Hz. Then, data was segmented from the beginning of each phase into 0.5 s long segments (240 duplets for the Random, 240 duplets for the long Structured, and 600 duplets for the short Structured). Segments containing samples with artefacts defined as bad data in more than 30% of the channels were rejected, and the remaining channels with artefacts were spatially interpolated.

### Neural entrainment per condition

The 0.5 s epochs belonging to the same condition were reshaped into non-overlapping epochs (*Benjamin et al., 2021*) of 7.5 s (15 duplets, 30 syllables), retaining the chronological order; thus, the timing of the steady-state response. Subjects who did not provide at least 50% artifact-free epochs for each condition (at least 8 long epochs during Random and 28 during Structured) were excluded from the entrainment analysis (32 included subjects in Experiment 1, and 32 included subjects in Experiment 2). The retained subjects for Experiment 1, on average provided 13.59 epochs for the Random condition (SD 2.07, range [8, 16]) and 48.16 for the Structured conditions (SD 5.89, range [33, 55]). The retained subjects for Experiment 2, on average provided 13.78 epochs for the Random condition (SD 1.93, range [8, 16]) and 46.88 for the Structured conditions (SD 5.62, range [36, 55]). After data rejection, data were referenced to the average and normalized by dividing by the standard deviation within an epoch across electrodes and time. Next, data were converted to the frequency domain using the Fast Fourier Transform (FFT) algorithm, and the ITC was estimated for each electrode during each condition (Random, Structured) as $ITC\,(f) = \frac{1}{N}\left|\sum_{i=1}^{N} e^{i\varphi\,(f,i)}\right|$, where N is the number of trials and φ(f,i) is the phase at frequency f and trial i. The ITC ranges from 0 to 1 (i.e. completely desynchronized activity to perfectly phased locked activity). Since we aim to detect an increase in signal synchronization at specific frequencies, the SNR was computed relative to the twelve adjacent frequency bins (six of each side corresponding to 0.8 Hz; *Kabdebon et al., 2022*). This procedure also enables correcting differences in the ITC due to a different number of trials. Specifically, the SNR was SNR(f) = (ITC(f)-mean(ITC$_{noise}$(f)))/std(ITC$_{noise}$(f)), where ITC$_{noise}$(f) is the ITC over the adjacent frequency bins. For statistical analysis, we compared the SNR at syllable rate (4 Hz) and duplet rate (2 Hz) against the average SNR over the 12 adjacent frequency bins using a one-tail paired permutation test (5000 permutations). We also directly compared the entrainment during the two conditions to individuate the electrodes showing a greater entrainment during the Structured than Random streams. We evaluated the interaction between Stream type (Random and Structured) and Familiarization type (Structured over Phonemes or Voices) by comparing the difference in entrainment between Structured and Random during the two experiments using a two sides unpaired permutation test (5000 permutations). All p-values were corrected across electrodes by FDR.

### Neural entrainment time course

The 0.5 s epochs were concatenated chronologically (2 mins of Random, 2 min of long Structured stream, and 5 min of short Structured blocks). The same analysis as above was performed in sliding time windows of 2 min with a 1 s step. A time window was considered valid if at least 8 out of the 16 epochs were free of motion artefacts. Missing values due to the presence of motion artifacts where linearly interpolated. Then, the entrainment time course at the syllable rate was computed as the average over the electrodes showing significant entrainment at 4 Hz during the Random condition, and at the duplet rate, as the average over the electrodes showing significant entrainment at 2 Hz during the Structured condition. Finally, data was smooth over a time window of 30 s.

To investigate the increase in the neural activity locked to the regularity during the long familiarisation, we fitted a LMM for each group of subjects. We included time as a fixed effect and random slopes and interceptions for individual subjects: $ITC \sim -1 + time + (1 + time|subject)$. We then compare the

time effect between groups by including all data in a single LMM with time and group as fixed effects: $ITC \sim -1 + time * group + (1 + time|subject)$.

## ERPs to test words

The pre-processed data were filtered between 0.2 and 20 Hz, and epoched between [–0.2, 2.0] s from the onset of the duplets. Epochs containing samples identified as artifacts by APICE procedure were rejected. Subjects who did not provide at least half of the trials (45 trials) per condition were excluded (34 subjects kept for Experiment 1, and 33 for Experiment 2). No subject was excluded based on this criterion in the Phoneme groups, and one subject was excluded in the Voice groups. For Experiment 1, we retained on average 77.47 trials (SD 9.98, range [52, 89]) for the Word condition and 77.12 trials (SD 10.04, range [56, 89]) for the Part-word condition. For Experiment 2, we retained on average 73.73 trials (SD 10.57, range [47, 90]) for the Word condition and 74.18 trials (SD 11.15, range [46, 90]) for the Part-word condition. Data were reference averaged and normalised within each epoch by dividing by the standard deviation across electrodes and time.

Since the grand average response across both groups and conditions returned to the pre-stimulus level at around 1500ms, we defined [0, 1500] ms as time windows of analysis. We first analysed the data using non-parametric cluster-based permutation analysis (*Oostenveld et al., 2011*) in the time window [0, 1500] ms (alpha threshold for clustering 0.10, neighbour distance ≤2.5 cm, clusters minimum size 3 and 5000 permutations).

We also analysed the data in seven ROIs to ensure that no other effects were present that were not caught by the cluster-based permutation analysis. By inspecting the grand average ERP across both experiments and conditions, we identified three characteristic topographies: (a) positivity over central electrodes, (b) positivity over frontal electrodes and negativity over occipital electrodes, and (c) positivity over prefrontal electrodes and negativity over temporal electrodes (*Figure 3—figure supplement 1*). Then, we defined seven symmetric ROIs: Central, Frontal Left, Frontal Right, Occipital, Prefrontal, Temporal Left, Temporal Right. We evaluated the main effect of Test-word type by comparing the EPRs between Words and Partwords (paired t-test) and the main effect of Familiarization type by comparing ERPs between Experiment 1 (structured over Phonemes) and Experiment 2 (structure over Voices; unpaired t-test). All p-values were FDR corrected by the number of time points (n=376) and ROIs (n=7). To test for possible interaction effects, we compared the difference between Words and Partwords between the two groups. To verify that the main effects were not driven by one condition or group, we computed the average on each of the time windows where a main effect was identified considering both the Test-word type and Familiarization type factors, and we ran a two ways-ANOVA (Test-word type x Familiarization type). Results are presented in *Figure 3—figure supplement 3*.

## Adults' behavioural experiment

57 French-speaking adults were tested in an online experiment analogous to the infant study through the Prolific platform. All participants provided informed consent and received monetary compensation for their participation. The study was approved by the Ethical Research Committee of Paris Saclay University under the reference CER-Paris-Saclay-2019–063. The same stimuli as in the infants' experiment were used. Participants first heard 2 min of familiarisation with the Structured stream. Then, they completed ten sessions of re-familiarisation and testing. Each re-familiarization lasted 30 s, and in each test session, all 18 test words were presented. The structure could be either over the phonetic or the voice content, and two lists were used (see *Appendix 1—table 2*). Participants were randomly assigned to one of the groups and to one list. The Phoneme group included 27 participants, and the Voice group 30 participants. Before starting the experiment, subjects were instructed to pay attention to an invented language because later, they would have to answer if different sequences adhered to the structure of the language. During the test phase, subjects were asked to scale their familiarity with each test word by clicking with a cursor on a scale from 1 to 6.

## Acknowledgements

We want to thank all the families who participated in the study. This research has received funding from the European Research Council (ERC) under the European Union's Horizon 2020 research and innovation program (grant agreement No. 695710).

# Additional information

## Funding

| Funder | Grant reference number | Author |
|---|---|---|
| Horizon 2020 Framework Programme | 695710 | Ghislaine Dehaene-Lambertz |

The funders had no role in study design, data collection and interpretation, or the decision to submit the work for publication.

## Author contributions

Ana Fló, Conceptualization, Data curation, Software, Formal analysis, Methodology, Writing – original draft, Writing – review and editing; Lucas Benjamin, Investigation, Writing – review and editing; Marie Palu, Investigation, Project administration; Ghislaine Dehaene-Lambertz, Conceptualization, Supervision, Funding acquisition, Writing – original draft, Project administration, Writing – review and editing

## Author ORCIDs

Ana Fló ⬤ https://orcid.org/0000-0002-3260-0559
Lucas Benjamin ⬤ https://orcid.org/0000-0002-9578-6039

## Ethics

Neonates were tested at the Port Royal Maternity (AP-HP), in Paris, France. Parents provided informed consent. The regional ethical committee for biomedical research (Comité de Protection des Personnes Region Centre Ouest 1, EudraCT/ID RCB: 2017-A00513-50) approved the protocol, and the study was carried out according to relevant guidelines and regulations.

Reviewer #1 (Public review): https://doi.org/10.7554/eLife.101802.3.sa1
Reviewer #2 (Public review): https://doi.org/10.7554/eLife.101802.3.sa2
Reviewer #3 (Public review): https://doi.org/10.7554/eLife.101802.3.sa3
Author response https://doi.org/10.7554/eLife.101802.3.sa4

# Additional files

## Supplementary files

MDAR checklist

## Data availability

This study involves sensitive data collected from human infants, which can only be used for scientific research and not for commercial applications. Parents have given consent for the data to be used strictly for scientific research. Consequently, individual infant data cannot be made publicly available but can be accessed on request. Researchers who wish to request access to the infant data must be affiliated with a public research institution and clearly outline the scientific objectives of their request, with replication being an acceptable goal. Requests should be submitted to Ghislaine Dehaene-Lambertz (gdehaene@gmail.com), and the ethical officer of the institution (CEA) will evaluate them. All the analysis tools for the infants' data and grand average results are publicly available. The adult data and analysis tools are publicly available. All openly available material can be found at https://osf.io/an4jk/.

The following dataset was generated:

| Author(s) | Year | Dataset title | Dataset URL | Database and Identifier |
|---|---|---|---|---|
| Benjamin L, Palu M, Dehaene-Lambertz G, Fló A | 2025 | Statistical learning beyond words in human neonates | https://osf.io/an4jk/ | Open Science Framework, an4jk |

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

# Appendix 1

## Post-hoc test: investigating female and male voice perception effects

Given that gender may be an important factor in infants' speech perception (newborns, for instance, prefer female voices at birth), we conducted tests to assess whether this dimension could have influenced the results observed in Experiment 2.

## Computation of TPs

We first quantified the transitional probabilities matrices during the structured stream of Experiment 2, considering that there were only two types of voices: Female and Male.

For List A, all transition probabilities were equal to 0.5 (P(M|F), P(F|M), P(M|M), P(F|F)), resulting in flat TPs throughout the stream (*Appendix 1—figure 1*, top). Therefore, we would not expect neural entrainment at the word rate (2 Hz) nor anticipate ERP differences between the presented duplets in the test phase.

For List B, P(M|F)=P(F|M)=0.66 while P(M|M)=P(F|F)=0.33, without a regular pattern of TP drops throughout the stream (*Appendix 1—figure 1* -, bottom). Although this pattern is unlikely to induce strong neural entrainment at 2 Hz, some degree of entrainment might have occasionally occurred due to some drops occurring at a 2 Hz frequency. Regarding the test phase, all three Words and only one Part-word presented alternating patterns (TP=0.6). Therefore, we cannot rule out that gender alternation might have affected the difference in ERPs between Words and Partwords in List B.

While it seems unlikely that gender alternation alone explains the entire pattern of results, as the effect was inconsistent and appeared in only one of the lists, we separately analysed the entrainment and ERP effects in each list to rule out this possibility

## Neural entrainment effect

We computed the average entrainment over the electrodes, which showed significant entrainment at 2 Hz (word rate). A comparison of the 2 Hz entrainment between participants who completed List A and List B showed no significant differences ($t(30) = -0.27$, $p = 0.79$). A test against zero for each list indicated significant entrainment in both cases (List A: $t(17) = 4.44$, $p = 0.00036$; List B: $t(13) = 3.16$, $p = 0.0075$). See *Appendix 1—figure 2*.

## ERP effect

We computed the mean activation within the time windows and electrodes of interest and compared the effects of word type and list using a two-way ANOVA. For the difference between 47 Words and Part-words over the positive cluster, we observed a main effect of word type ($F(1,31) = 5.902$, $p = 0.021$), with no effects of list or interactions (ps > 0.1). Over the negative cluster, we again observed a main effect of word type ($F(1,31) = 10.916$, $p = 0.0016$), with no effects of list or interactions (ps > 0.1). See *Appendix 1—figure 3*.

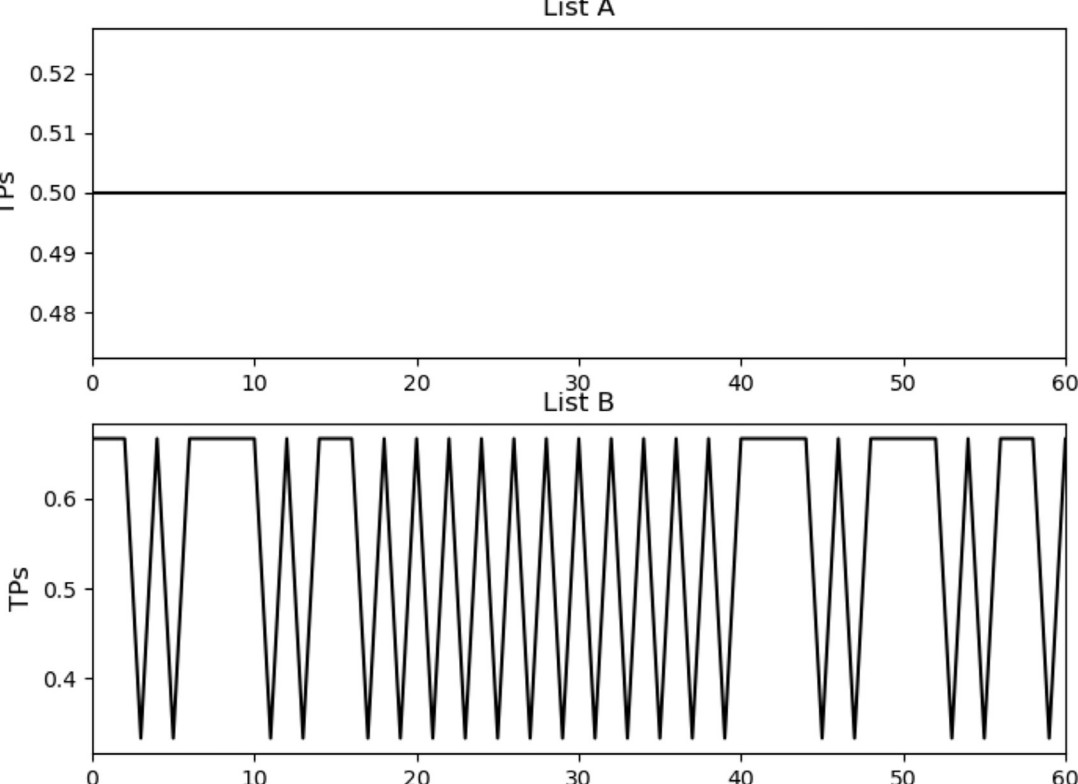

**Appendix 1—figure 1.** Transition probabilities (TPs) across the structured stream in Experiment 2, considering voices processed by gender (Female or Male). Top: List A. Bottom: List B.

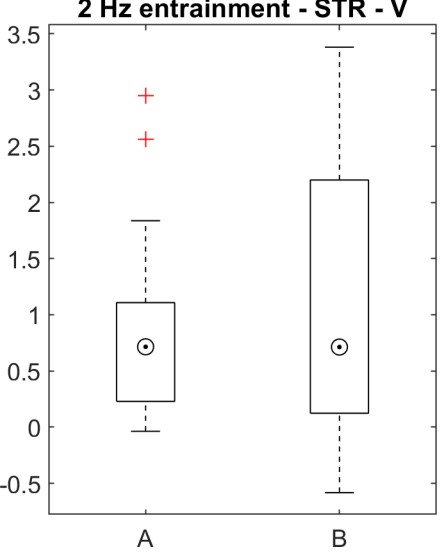

**Appendix 1—figure 2.** Neural entrainment at word rate (2Hz) during the structured stream of Experiment 2 for Lists A and B.

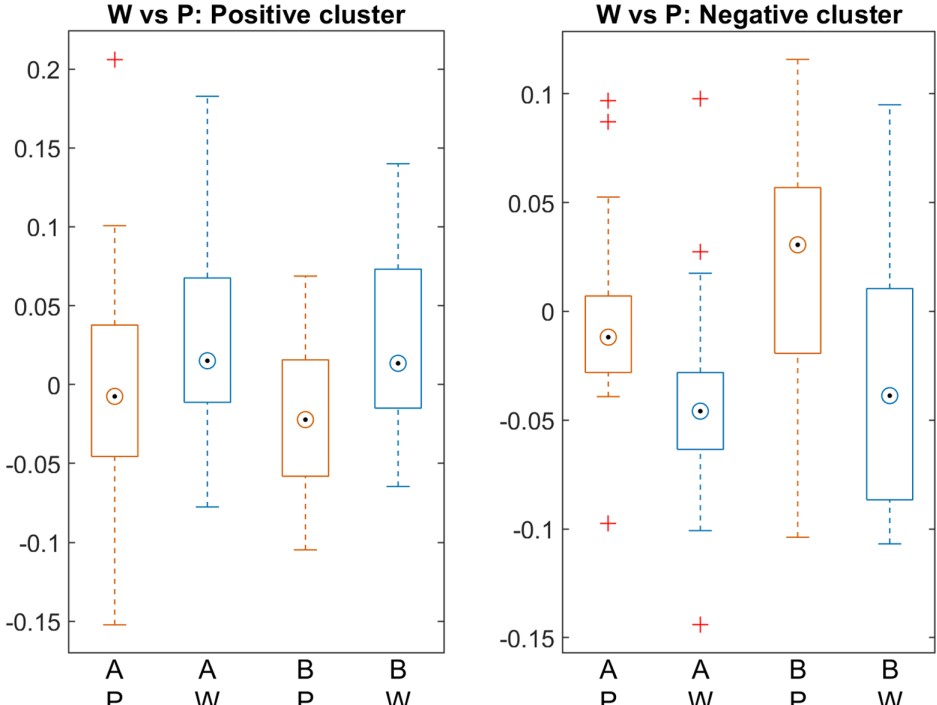

**Appendix 1—figure 3.** Difference in ERP voltage (Words – Part-words) for the twPo lists (A and B); W=Words; P=Part-Words.

**Appendix 1—table 1.** Voice stimuli.
properties of the six voices used to construct the stimuli. The Voice column indicates the name of the voice from the MBROLA diphone database voice used.

|  | Voice | Gender | Pitch (Hz) |
|---|---|---|---|
| Ma | fr3 | Male | 75 |
| Mb | fr1 | Male | 108 |
| Mc | fr7 | Male | 140 |
| Fa | fr2 | Female | 133 |
| Fb | it4 | Female | 190 |
| Fc | fr4 | Female | 247 |

**Appendix 1—table 2.** Stimuli.
Words and Part-words for each experiment and list.

| Experiment 1 | | | | Experiment 2 | | | |
|---|---|---|---|---|---|---|---|
| List A | | List B | | List A | | List B | |
| Word | Part-word | Word | Part-word | Word | Part-word | Word | Part-word |
| kida | dape | dape | kida | FbMb | MbFc | MbFc | FbMb |
| petu | tubo | tubo | petu | FcFa | FaMa | FaMa | FcFa |
| bogɛ | gɛki | gɛki | bogɛ | MaMc | McFb | McFb | MaMc |

