## [Editor Report · eLife Assessment]

The manuscript provides **important** new insights into the mechanisms of statistical learning in early human development, showing that statistical learning in neonates occurs robustly and is not limited to linguistic features but occurs across different domains. The evidence is **convincing** and the findings are highly relevant for researchers working in several domains, including developmental cognitive neuroscience, developmental psychology, linguistics, and speech pathology.

---

## [Referee Report · Reviewer #1 (Public review)]

Summary:

Parsing speech into meaningful linguistic units is a fundamental yet challenging task that infants face while acquiring the native language. Computing transitional probabilities (TPs) between syllables is a segmentation cue well-attested since birth. In this research, the authors examine whether newborns compute TPs over any available speech feature (linguistic and non-linguistic), or whether by contrast newborns favor computation of TPs over linguistic content over non-linguistic speech features such as speaker voice. Using EEG and the artificial language learning paradigm, they record the neural responses of two groups of newborns presented with speech streams in which either phonetic content or speaker voice are structured to provide TPs informative of word boundaries, while the other dimension provides uninformative information. They compare newborns' neural responses to these structured streams to their processing of a stream in which both dimensions vary randomly. After the random and structured familiarization streams, the newborns are presented with (pseudo)words as defined by their informative TPs, as well as partwords (that is, sequences that straddle a word boundary), extracted from the same streams. Analysis of the neural responses show that while newborns neural activity entrained to the syllabic rate (2 Hz) when listening to the random and structured streams, it additionally entrained at the word rate (4 Hz) only when listening to the structured streams, finding no differential response between the streams structured around voice or phonetic information. Newborns showed also different neural activity in response to the words and part words. In sum, the study reveals that newborns compute TPs over linguistic and non-linguistic features of speech, these are calculated independently, and linguistic features do not lead to a processing advantage.

Strengths:

This interesting research furthers our knowledge of the scope of the statistical learning mechanism, which is confirmed to be a general-purpose powerful tool that allows humans to extract patterns of co-occurring events while revealing no apparent preferential processing for linguistic features. To answer its question, the study combines a highly replicated and well-established paradigm, i.e. the use of an artificial language in which pseudowords are concatenated to yield informative TPs to word boundaries, with a state-of-the-art EEG analysis, i.e. neural entrainment. The sample size of the groups is sufficient to ensure power, and the design and analysis are solid and have been successfully employed before.

Weaknesses:

There are no significant weaknesses to signal in the manuscript. However, in order to fully conclude that there is no obvious advantage for the linguistic dimension in neonates, future studies should pit both dimensions against each other, to determine whether statistical learning weighs linguistic and non-linguistic features equally, or whether phonetic content is preferentially processed.

To sum up, the authors achieved their central aim of determining whether TPs are computed over both linguistic and non-linguistic features, and their conclusions are supported by the results. This research is important for researchers working on language and cognitive development, and language processing, as well as for those working on cross-species comparative approaches.

Comments on revisions:

The authors have addressed my suggestions. I have no further comments.

---

## [Referee Report · Reviewer #2 (Public review)]

Summary:

The manuscript investigates to what degree neonates show evidence for statistical learning from regularities in streams of syllables, either with respect to phonemes or with respect to speaker identity. Using EEG, the authors found evidence for both, stronger entrainment to regularities as well as ERP differences in response to violations of previously introduced regularities. In addition, violations of phoneme regularities elicited an ERP pattern which the authors argue might index a precursor of the N400 response in older children and adults.

Strengths:

All in all, this is a very convincing paper, which uses a clever manipulation of syllable streams to target the processing of different features. The combination of neural entrainment and ERP analysis allows for the assessment of different processing stages, and implementing this paradigm in a comparably large sample of neonates is impressive.

Weaknesses

The authors addressed all the concerns I previously raised well and I have no further comments.

---

## [Referee Report · Reviewer #3 (Public review)]

Summary:

This study is focused on testing whether statistical learning (a mechanism for parsing the speech signal into smaller chunks) preferentially operates over certain features of the speech at birth in humans. The features under investigation are phonetic content and speaker identity. Newborns are tested in an EEG paradigm in which they are exposed to a long stream of syllables. In Experiment 1, newborns are familiarized with a sound stream that comprises regularities (transitional probabilities) over syllables (e.g., "pe" followed by "tu" in "petu" with 1.0 probability) while the voices uttering the syllables remain random. In Experiment 2, newborns are familiarized with the same sound stream but, this time, the regularities are built over voices (e.g., "green voice" followed by "red voice" with 1.0 probability) while the concatenation of syllables stays random. At the test, all newborns listened to duplets (individual chunks) that either matched or violated the structure of the familiarization. In both experiments, newborns showed neural entrainment to the regularities implemented in the stream, but only the duplets defined by transitional probabilities over syllables (aka word forms) elicited a N400 ERP component. These results suggest that statistical learning operates in parallel and independently on different dimensions of the speech already at birth and that there seems to be an advantage for processing statistics defining word forms rather than voice patterns.

Strengths:

This paper presents an original experimental design that combines two types of statistical regularities in a speech input. The design is robust and appropriate for EEG with newborns. I appreciated the clarity of the Methods section. There is also a behavioral experiment with adults that acts like a control study for newborns. The research question is interesting, and the results add new information about how statistical learning works at the beginning of postnatal life, and on which features of the speech. The figures are clear and helpful in understanding the methods, especially the stimuli and how the regularities were implemented.

Weaknesses:

I appreciated how the authors addressed my previous comments and concerns. I am satisfied with the changes made by the authors. I believe the paper reads much better. Also, the adjustment to the theoretical framework suits well.

---

## [Author Response]

The following is the authors’ response to the original reviews.

We thank the three reviewers for their positive comments and useful suggestions. We have implemented most of the reviewers’ recommendations and hope the manuscript is clearer now.

The main modifications are:

- A revision of the introduction to better explain what Transitional Probabilities are and clarify the rationale of the experimental design

- A revision of the discussion

- To tune down and better explain the interpretation of the different responses between duplets after a stream with phonetic or voice regularities (possibly an N400).

- To better clarify the framing of statistical learning as a universal learning mechanism that might share computational principles across features (or domains).

Below, we provide detailed answers to each reviewer's point.

**Response to Reviewer 1:**
There are no significant weaknesses to signal in the manuscript. However, in order to fully conclude that there is no obvious advantage for the linguistic dimension in neonates, it would have been most useful to test a third condition in which the two dimensions were pitted against each other, that is, in which they provide conflicting information as to the boundaries of the words comprised in the artificial language.This last condition would have allowed us to determine whether statistical learning weighs linguistic and non-linguistic features equally, or whether phonetic content is preferentially processed.

We appreciate the reviewers' suggestion that a stream with conflicting information would provide valuable insights. In the present study, we started with a simpler case involving two orthogonal features (i.e., phonemes and voices), with one feature being informative and the other uninformative, and we found similar learning capacities for both. Future work should explore whether infants—and humans more broadly—can simultaneously track regularities in multiple speech features. However, creating a stream with two conflicting statistical structures is challenging. To use neural entrainment, the two features must lead to segmentation at different chunk sizes so that their effects lead to changes in power/PLV at different frequencies—for instance, using duplets for the voice dimension and triplets for the linguistic dimension (or vice versa). Consequently, the two dimensions would not be directly comparable within the same participant in terms of the number of distinguishable syllables/voices, memory demand, or SNR given the 1/F decrease in amplitude of background EEG activity. This would involve comparisons between two distinct groups counter-balancing chunk size and linguistic non-linguistic dimension. Considering the test phase, words for one dimension would have been part-words for the other dimension. As we are measuring differences and not preferences, interpreting the results would also have been difficult. Additionally, it may be difficult to find a sufficient number of clearly discriminable voices for such a design (triplets imply 12 voices). Therefore, an entirely different experimental paradigm would need to be developed.

If such a design were tested, one possibility is that the regularities for the two dimensions are calculated in parallel, in line with the idea that the calculation of statistical regularities is a ubiquitous implicit mechanism (see Benjamin et al., 2024, for a proposed neural mechanism). Yet, similar to our present study, possibly only phonetic features would be used as word candidates. Another possibility is that only one informative feature would be explicitly processed at a time due to the serial nature of perceptual awareness, which may prioritise one feature over the other.

We added one sentence in the discussion stating that more research is needed to understand whether infants can track both regularities simultaneously (p.13, l.270 “Future work could explore whether they can simultaneously track multiple regularities.”).

Note: The reviewer’s summary contains a typo: syllabic rate (4 Hz) –not 2 Hz, and word rate (2 Hz) –not 4 Hz.

**Response to Reviewer 2:**
N400: I am skeptical regarding the interpretation of the phoneme-specific ERP effect as a precursor of the N400 and would suggest toning it down. While the authors are correct in that infant ERP components are typically slower and more posterior compared to adult components, and the observed pattern is hence consistent with an adult N400, at the same time, it could also be a lot of other things. On a functional level, I can't follow the author's argument as to why a violation in phoneme regularity should elicit an N400, since there is no evidence for any semantic processing involved. In sum, I think there is just not enough evidence from the present paradigm to confidently call it an N400.

The reviewer is correct that we cannot definitively determine the type of processing reflected by the ERP component that appears when neonates hear a duplet after exposure to a stream with phonetic regularities. We interpreted this component as a precursor to the N400, based on prior findings in speech segmentation tasks without semantic content, where a ~400 ms component emerged when adult participants recognised pseudowords (Sander et al., 2002) or during structured streams of syllables (Cunillera et al., 2006, 2009). Additionally, the component we observed had a similar topography and timing to those labelled as N400 in infant studies, where semantic processing was involved (Parise et al., 2010; Friedrich & Friederici, 2011).

Given our experimental design, the difference we observed must be related to the type of regularity during familiarisation (either phonemes or voices). Thus, we interpreted this component as reflecting lexical search— a process which could be triggered by a linguistic structure but which would not be relevant to a non-linguistic regularity such as voices. However, we are open to alternative interpretations. In any case, this difference between the two streams reveals that computing regularities based on phonemes versus voices does not lead to the same processes.

We revised the abstract (p.2, l.33) and the discussion of this result (p.15, l.299), toning them down. We hope the rationale of the interpretation is clearer now, as is the fact that it is just one possible interpretation of the results.

Female and male voices: Why did the authors choose to include male and female voices? While using both female and male stimuli of course leads to a higher generalizability, it also introduces a second dimension for one feature that is not present for this other (i.e., phoneme for Experiment 1 and voice identity plus gender for Experiment 2). Hence, couldn't it also be that the infants extracted the regularity with which one gender voice followed the other? For instance, in List B, in the words, one gender is always followed by the other (M-F or F-M), while in 2/3 of the part-words, the gender is repeated (F-F and M-M). Wouldn't you expect the same pattern of results if infants learned regularities based on gender rather than identity?

We used three female and three male voices to maximise acoustic variability. The streams were synthesised using MBROLA, which provides a limited set of artificial voices. Indeed, there were not enough French voices of acceptable quality, so we also used two Italian voices (the phonemes used existed in both Italian and French).

Voices differ in timbre, and female voices tend to be higher pitched. However, it is sometimes difficult to categorise low-pitched female voices and high-pitched male voices. Given that gender may be an important factor in infants' speech perception (newborns, for instance, prefer female voices at birth), we conducted tests to assess whether this dimension could have influenced our results.

We report these analyses in SI and referred to them in the methods section (p.25, l.468 “We performed post-hoc tests to ensure that the results were not driven by a perception of two voices: female and male (see SI).”).

We first quantified the transitional probabilities matrices during the structured stream of Experiment 2, considering that there are only two types of voices: Female and Male.

For List A, all transition probabilities are equal to 0.5 (P(M|F), P(F|M), P(M|M), P(F|F)), resulting in flat TPs throughout the stream (see Author response image 1, top). Therefore, we would not expect neural entrainment at the word rate (2 Hz), nor would we anticipate ERP differences between the presented duplets in the test phase.

For List B, P(M|F)=P(F|M)=0.66 while P(M|M)=P(F|F)=0.33. However, this does not produce a regular pattern of TP drops throughout the stream (see Author response image 1, bottom). As a result, strong neural entrainment at 2 Hz was unlikely, although some degree of entrainment might have occasionally occurred due to some drops occurring at a 2 Hz frequency. Regarding the test phase, all three Words and only one Part-word presented alternating patterns (TP=0.6). Therefore, the difference in the ERPs between Words and Part- words in List B might be attributed to gender alternation.

However, it seems unlikely that gender alternation alone explains the entire pattern of results, as the effect is inconsistent and appears in only one of the lists. To rule out this possibility, we analysed the effects in each list separately.

**Author response image 1. sa4fig1:** Transition probabilities (TPs) across the structured stream in Experiment 2, considering voices processed by gender (Female or Male). Top: List A. Bottom: List B.

We computed the mean activation within the time windows and electrodes of interest and compared the effects of word type and list using a two-way ANOVA. For the difference between Words and Part-words over the positive cluster, we observed a main effect of word type (F(1,31) = 5.902, p = 0.021), with no effects of list or interactions (p > 0.1). Over the negative cluster, we again observed a main effect of word type (F(1,31) = 10.916, p = 0.0016), with no effects of list or interactions (p > 0.1). See Author response image 2.

**Author response image 2. sa4fig2:** Difference in ERP voltage (Words – Part-words) for the two lists (A and B); W=Words; P=Part-Words.

We conducted a similar analysis for neural entrainment during the structured stream on voices. A comparison of entrainment at 2 Hz between participants who completed List A and List B showed no significant differences (t(30) = -0.27, p = 0.79). A test against zero for each list indicated significant entrainment in both cases (List A: t(17) = 4.44, p = 0.00036; List B: t(13) = 3.16, p = 0.0075). See Author response image 3.

**Author response image 3. sa4fig3:** Neural entrainment at 2Hz during the structured stream of Experiment 2 for Lists A and B.

Words entrainment over occipital electrodes: Do you have any idea why the duplet entrainment effect occurs over the electrodes it does, in particular over the occipital electrodes (which seems a bit unintuitive given that this is a purely auditory experiment with sleeping neonates).

Neural entrainment might be considered as a succession of evoked response induced by the stream. After applying an average reference in high-density EEG recordings, the auditory ERP in neonates typically consists of a central positivity and a posterior negativity with a source located at the electrical zero in a single-dipole model i.e. approximately in the superior temporal region (Dehaene-Lambertz & Dehaene, 1994). In adults, because of the average reference (i.e. the sum of voltages is equal to zero at each time point) and because the electrodes cannot capture the negative pole of the auditory response, the negativity is distributed around the head. In infants, however, the brain is higher within the skull, allowing for a more accurate recording of the negative pole of the auditory ERP (see Figure 4 for the location of electrodes in an infant head model).

Besides the posterior electrodes, we can see some entrainment on more anterior electrodes that probably corresponds to the positive pole of the auditory ERP.

We added a phrase in the discussion to explain why we can expect phase-locked activity in posterior electrodes (p.14, l.277: “Auditory ERPs, after reference-averaged, typically consist of a central positivity and posterior negativity”).

**Author response image 4. sa4fig4:** International 10–20 sensors' location on the skull of an infant template, with the underlying 3-D reconstruction of the grey-white matter interface and projection of each electrode to the cortex. Computed across 16 infants (from Kabdebon et al, Neuroimage, 2014). The O1, O2, T5, and T6 electrodes project lower than in adults.

**Response to Reviewer 3:**
(1) While it's true that voice is not essential for language (i.e., sign languages are implemented over gestures; the use of voices to produce non-linguistic sounds, like laughter), it is a feature of spoken languages. Thus I'm not sure if we can really consider this study as a comparison between linguistic and non-linguistic dimensions. In turn, I'm not sure that these results show that statistical learning at birth operates on non-linguistic features, being voices a linguistic dimension at least in spoken languages. I'd like to hear the authors' opinions on this.

On one hand, it has been shown that statistical learning (SL) operates across multiple modalities and domains in human adults and animals. On the other hand, SL is considered essential for infants to begin parsing speech. Therefore, we aimed to investigate whether SL capacities at birth are more effective on linguistic dimensions of speech, potentially as a way to promote language learning.

We agree with the reviewer that voices play an important role in communication (e.g., for identifying who is speaking); however, they do not contribute to language structure or meaning, and listeners are expected to normalize across voices to accurately perceive phonemes and words. Thus, voices are speech features but not linguistic features. Additionally, in natural speech, there are no abrupt voice changes within a word as in our experiment; instead, voice changes typically occur on a longer timescale and involve only a limited number of voices, such as in a dialogue. Therefore, computing regularities based on voice changes would not be useful in real-life language learning. We considered that contrasting syllables and voices was an elegant way to test SL beyond its linguistic dimension, as the experimental paradigm is identical in both experiments.

We have rephrased the introduction to make this point clearer. See p.5, l.88-92: “To test this, we have taken advantage of the fact that syllables convey two important pieces of information for humans: what is being said and who is speaking, i.e. linguistic content and speaker’s identity. While statistical learning…”.

Along the same line, in the Discussion section, the present results are interpreted within a theoretical framework showing statistical learning in auditory non-linguistic (string of tones, music) and visual domains as well as visual and other animal species. I'm not sure if that theoretical framework is the right fit for the present results.(2) I'm not sure whether the fact that we see parallel and independent tracking of statistics in the two dimensions of speech at birth indicates that newborns would be able to do so in all the other dimensions of the speech. If so, what other dimensions are the authors referring to?

The reviewer is correct that demonstrating the universality of SL requires testing additional modalities and acoustic dimensions. However, we postulate that SL is grounded in a basic mechanism of long-term associative learning, as proposed in Benjamin et al. (2024), which relies on a slow decay in the representation of a given event. This simple mechanism, capable of operating on any representational output, accounts for many types of sequence learning reported in the literature (Benjamin et al., in preparation).

We have revised the discussion to clarify this theoretical framework.

In p.13, l.264: “This mechanism might be rooted in associative learning processes relying on the co- existence of event representations driven by slow activation decays (Benjamin et al., 2024). ”

In p., l. 364: “Altogether, our results show that statistical learning works similarly on different speech features in human neonates with no clear advantage for computing linguistically relevant regularities in speech. This supports the idea that statistical learning is a general learning mechanism, probably operating on common computational principles across neural networks (Benjamin et al., 2024)…”.

(3) Lines 341-345: Statistical learning is an evolutionary ancient learning mechanism but I do not think that the present results are showing it. This is a study on human neonates and adults, there are no other animal species involved therefore I do not see a connection with the evolutionary history of statistical learning. It would be much more interesting to make claims on the ontogeny (rather than philogeny) of statistical learning, and what regularities newborns are able to detect right after birth. I believe that this is one of the strengths of this work.

We did not intend to make claims about the phylogeny of SL. Since SL appears to be a learning mechanism shared across species, we use it as a framework to suggest that SL may arise from general operational principles applicable to diverse neural networks. Thus, while it is highly useful for language acquisition, it is not specific to it.

We have removed the sentence “Statistical learning is an evolutionary ancient learning mechanism.”, and replaced it by (p.18, l.364) “Altogether, our results show that statistical learning works similarly on different speech features in human neonates with no clear advantage for computing linguistically relevant regularities in speech.” We now emphasise in the discussion that infants compute regularities on both features and propose that SL might be a universal learning mechanism sharing computational principles (Benjamin et al., 2024) (see point 2).

(4) The description of the stimuli in Lines 110-113 is a bit confusing. In Experiment 1, e.g., "pe" and "tu" are both uttered by the same voice, correct? ("random voice each time" is confusing). Whereas in Experiment 2, e.g., "pe" and "tu" are uttered by different voices, for example, "pe" by yellow voice and "tu" by red voice. If this is correct, then I recommend the authors to rephrase this section to make it more clear.

To clarify, in Experiment 1, the voices were randomly assigned to each syllable, with the constraint that no voice was repeated consecutively. This means that syllables within the same word were spoken by different voices, and each syllable was heard with various voices throughout the stream. As a result, neonates had to retrieve the words based solely on syllabic patterns, without relying on consistent voice associations or specific voice relationships.

In Experiment 2, the design was orthogonal: while the syllables were presented in a random order, the voices followed a structured pattern. Similar to Experiment 1, each syllable (e.g., “pe” and “tu”) was spoken by different voices. The key difference is that in Experiment 2, the structured regularities were applied to the voices rather than the syllables. In other words, the “green” voice was always followed by the “red” voice for example but uttered different syllables.

We have revised the description of the stimuli and the legend of Figure 1 to clarify these important points.

See p.6, l. 113: “The structure consisted of the random concatenation of three duplets (i.e., two-syllable units) defined only by one of the two dimensions. For example, in Experiment 1, one duplet could be petu with each syllable uttered by a random voice each time they appear in the stream (e.g pe is produced by voice1 and tu by voice6 in one instance and in another instance pe is produced by voice3 and tu by voice2). In contrast, in Experiment 2, one duplet could be the combination [voice1- voice6], each uttering randomly any of the syllables.”

p.20, l. 390 (Figure 1 legend): “For example, the two syllables of the word “petu” were produced by different voices, which randomly changed at each presentation of the word (e.g. “yellow” voice and “green” voice for the first instance, “blue” and “purple” voice for the second instance, etc..). In Experiment 2, the statistical structure was based on voices (TPs alternated between 1 and 0.5), while the syllables changed randomly (uniform TPs of 0.2). For example, the “green” voice was always followed by the “red” voice, but they were randomly saying different syllables “boda” in the first instance, “tupe” in the second instance, etc... “

(5) Line 114: the sentence "they should compute a 36 x 36 TPs matrix relating each acoustic signal, with TPs alternating between 1/6 within words and 1/12 between words" is confusing as it seems like there are different acoustic signals. Can the authors clarify this point?

Thank you for highlighting this point. To clarify, our suggestion is that neonates might not track regularities between phonemes and voices as separate features. Instead, they may treat each syllable-voice combination as a distinct item—for example, "pe" spoken by the "yellow" voice is one item, while "pe" spoken by the "red" voice is another. Under this scenario, there would be a total of 36 unique items (6 syllables × 6 voices), and infants would need to track regularities between these 36 combinations.

We have modified this sentence in the manuscript to make it clearer.

See p.7, l. 120: “If infants at birth compute regularities based on a neural representation of the syllable as a whole, i.e. comprising both phonetic and voice content, this would require computing a 36 × 36 TPs matrix relating each token.”

**Reviewer #1 (Recommendations for the authors):**

(1) The acronym TP should be spelled out, and a brief description of the fact that dips in TPs signal boundaries while high TPs signal a cohesive unit could be useful for non-specialist readers.

We have added it at the beginning of the introduction (lines 52-60)

(2) p.5, l.76: "Here, we aimed to further characterise the characteristics of this mechanism...". I suggest this is rephrased as "to further characterise this mechanism".

We have changed it as suggested by the reviewer (now p.5, l.81)

(3) p.9, l.172: "[...] this contribution is unlikely since the electrodes differ from the electrodes, showing enhanced word-rate activity at 2 Hz."It is unclear which electrodes differ from which electrodes. I figure that the authors mean that the electrodes showing stronger activity at 2 Hz differ from those showing it at 4 Hz, but the sentence could use rephrasing.

This part has been rephrased (p.9, l.177-181)

(4) p.10, l.182: "[...] the entrainment during the first minute of the structure stream […]".Structured stream.

It has been corrected (p.10, l.190)

(5) p.12, l.234: "we compared STATISTICAL LEARNING"Why the use of capitals?

This was an error and it was corrected (p.12, l.242).

(6) p.15, l.298: "[...] suggesting that such negativity might be related to semantic."The sentence feels incomplete. To semantics? To the processing of semantic information?

The phrase has been corrected (p.15, l.314). Additionally, the discussion of the posterior negativity observed for duplets after familiarisation with a stream with regularities over phonemes has been rephrased (p.15, l.)

(7) Same page, l.301: "3-mo-olds" 3-month-olds.

It has been corrected (now in p.16, l.333)

(8) Same page, l.307: "see also (Bergelson and Aslin, 2017)" (see also Bergelson and Aslin, 2017).

It has been corrected (now in p.17, l.340)

(9) Same page, l.310: "[...] would be considered as possible candidate" As possible candidates.

This has been rephrased and corrected (now in p.17, l.343)

**Reviewer #2 (Recommendations for the authors):**

(1) Figure 2: The authors mention a "thick orange line", which I think should be a "thick black line".

We are sorry for this. It has been corrected.

(2) Ln 166: Should be Figure 2C rather than 3C.

It has been corrected (now in p.9, l.173)

(3) Figure 4 is not referenced in the manuscript.

We referred to it now on p. 12, l.236